# Valorization of the Photo-Protective Potential of the Phytochemically Standardized Olive (*Olea europaea* L.) Leaf Extract in UVA-Irradiated Human Skin Fibroblasts

**DOI:** 10.3390/molecules27165144

**Published:** 2022-08-12

**Authors:** Paulina Machała, Oleksandra Liudvytska, Agnieszka Kicel, Angela Dziedzic, Monika A. Olszewska, Halina Małgorzata Żbikowska

**Affiliations:** 1Department of General Biochemistry, Faculty of Biology and Environmental Protection, University of Lodz, 141/143 Pomorska, 90-236 Lodz, Poland; 2Department of Pharmacognosy, Faculty of Pharmacy, Medical University of Lodz, 1 Muszynskiego, 90-151 Lodz, Poland

**Keywords:** human skin fibroblast, oleuropein, olive leaf extract, UVA, caspases, ROS, thioredoxin reductase, interleukin-2

## Abstract

Leaves of *Olea europaea* are a by-product of the olive oil industry and a dietary supplement with acknowledged antioxidant and anti-inflammatory activity but underestimated photoprotective potential. We investigated the protective effects of the LC-PDA-MS/MS standardized ethanol-water extract of olive leaves (OLE), containing 26.2% total phenols and 22.2% oleuropein, with underlying mechanisms against the UVA-induced oxidative damage in human dermal fibroblasts. Hs68 cells were pre-treated (24 h) with OLE (2.5–25 μg/mL) or the reference antioxidants, quercetin and ascorbic acid (25 μg/mL), followed by irradiation (8 J/cm^2^). OLE significantly reduced the UVA-induced DNA damage and reactive oxygen species (ROS) overproduction and increased the thioredoxin reductase (TrxR) expression and post-radiation viability of fibroblasts by inhibiting their apoptosis. Both intrinsic and extrinsic apoptotic signaling pathways appeared to be inhibited by OLE, but the activity of caspase 9 was the most reduced. We hypothesized that the TrxR up-regulation by OLE could have prevented the UVA-induced apoptosis of Hs68 cells. In addition, a significant decrease in UVA-induced secretion levels of tumor necrosis factor (TNF-α) and interleukin-2 (IL-2) was shown in human lymphocyte culture in response to OLE treatment. In summary, our results support the beneficial effect of OLE in an in vitro model and indicate its great potential for use in the cosmetic and pharmaceutical industry as a topical photoprotective, antioxidant, and anti-inflammatory agent.

## 1. Introduction

Overexposure to ultraviolet (UV) radiation from the sun or artificial sources causes a range of acute and delayed skin complications. The former involves, i.e., erythema (sunburn), edema and inflammation, while the latter includes premature skin ageing (photoaging) and skin disorders such as hyperpigmentation and cancers.

UVA radiation (320–400 nm) is the predominant UV band that reaches the earth’s surface. UVA penetrates the skin deeper than UVB; therefore it can interact with dermal fibroblasts, connective tissues and blood vessels in the dermis skin layer [1]. UVA can cause DNA damage mainly indirectly, via the excessive production of reactive oxygen species (ROS), but also directly, by inducing the formation of cyclobutane pyrimidine dimers (CPDs) [2]. UVA-induced ROS and irreparable DNA damage initiate several signal transduction cascades, such as activator protein-1 (AP-1) and nuclear factor- kappa B (NF-kB) to release proinflammatory cytokines [3]. In addition, ROS generation leads to oxidative damage of other cellular macromolecules and structures (e.g., proteins, membranes) which can trigger cell death, most commonly by apoptosis. The topical application of sunscreens along with endogenous (oral) antioxidant photoprotection is a general strategy to protect skin cells against the deleterious effect of UV radiation [4]. Unfortunately, the components of traditional sunscreens may cause skin irritation or allergic reactions [5,6,7], mostly have a short half-life on the skin [8] and may adversely impact the environment [9,10]. To avoid or reduce these side effects, the currently used sunscreens are increasingly enriched with plant compounds containing numerous bioactive substances, including antioxidants.

The olive tree (*Olea europaea* L.) leaves have long been used in folk medicine and as a herbal tea in Europe and the Mediterranean countries [11]. They are one of the primary olive oil industry by-products and a cheap raw material that can be used as an effective source of high-added-value products [12] such as olive leaf extract (OLE). The leaves and OLE have been marketed as dietary supplements for multiple health benefits, particularly a broad antioxidant activity, which results in potent cardioprotective and chemoprotective effects [13,14]. They were also beneficial in hepatic disorders, obesity, and diabetes [15]. The olive leaf-based products are available on the market as whole dried leaves, leaf powder, and leaf extract in various pharmaceutical formulations, mostly capsules [16]. The standardized OLE is especially valuable as it contains concentrated active components. Several compounds have been found in the extract, such as simple phenols, flavonoids and especially secoiridoids, which are primarily responsible for its therapeutic activities [15]. Oleuropein is the main constituent of OLE, and the demonstrated biological activities of the extract are believed to largely depend on its content. Oleuropein, a (3,4-dihydroxyphenyl) ethanol (hydroxytyrosol) ester with α-glucosylated elenolic acid, is one of the secoiridoid olive metabolites. The main demonstrated biological activities of oleuropein are antioxidant and anti-inflammatory effects, but it has also been reported as an antimicrobial, antiviral, antiatherogenic, antihypertensive, and anti-clastogenic agent [15,17].

The previous mechanistic in vitro and in vivo studies of OLE and its main components focused mainly on their health benefits within the cardiovascular system [12,18,19], while their dermato-protective effects were sparsely explored. In particular, OLE has been reported to inhibit skin damage caused by acute UVB irradiation in mice [11], while hydroxytyrosol, a structural unit of the oleuropein molecule, has been found to protect DNA strand breaking and inhibit the pro-inflammatory response of UVB-irradiated human skin keratinocytes in vitro [20,21]. However, the effects of OLE on normal human dermal cells and UVA-induced cellular damage are so far unknown.

Therefore, in this study we evaluated the influence of OLE (the commercial ethanol-water extract of *O. europaea* leaves, standardized to 20% oleuropein content) on viability, apoptosis, ROS production, DNA damage, and thioredoxin reductase (TrxR) expression in UVA-irradiated human dermal fibroblasts (Hs68). In addition, the anti-inflammatory effect of OLE was examined by measuring the concanavalin A (con A) or UVA-induced secretion of tumor necrosis factor α (TNF-α) and interleukin-2 (IL-2) in human peripheral blood mononuclear cells (PBMCs). The quality and standardization parameters of OLE were confirmed by thorough phytochemical profiling, including LC-MS/MS structural identification of primary extract constituents. Our results demonstrate that the standardized OLE significantly protects human dermal cells against UVA-induced oxidative damage, prevents UV-induced apoptotic fibroblast death through TrxR up-regulation, displays anti-inflammatory activity, and is thus a promising photoprotective agent.

## 2. Results and Discussion

### 2.1. Identification of Bioactive Compounds in O. europaea Leaf Extract

Dietary supplements containing plant extracts constitute a strongly developing branch of the food and pharmaceutical industries, but their quality is often questioned due to the insufficient regulatory jurisdiction [22]. Therefore, the quality parameters of the *O. europaea* leaf extract (OLE), a commercial dietary supplement and test material of the present study, were verified in the thorough phytochemical profiling. The qualitative LC-PDA-MS^3^ analysis of OLE enabled the separation and complete or tentative identification of 22 constituents (UHPLC peaks 1–22, Figure 1, Table 1) belonging to several classes of phytochemicals such as secoiridoids, phenylethanoids and flavonoids, typical for the olive leaves [20,21,22,23,24,25,26], which confirmed the authentication of the extract. The structural identity of phytocompounds was established by comparing their retention times, UV–Vis spectral data and MS/MS profiles with literature data or those recorded for authentic standards.

#### 2.1.1. Secoiridoids

As shown in Figure 1, OLE was rich in secoiridoids, especially oleosides, which are characteristic compounds for the Oleaceae family [23,24]. In accordance with literature data [24,25], the dominant constituent of the extract (**19**) was identified as oleuropein by comparison with the reference standard. Moreover, the MS fragmentation pattern of compound **19** corresponded to that reported in the literature for oleuropein [26,27,28,29] and revealed the deprotonated ion [M − H]^−^ at *m*/*z* 539 and MS^2^ fragment ions at *m*/*z* 377, 345, 307 and 275. The typical MS^2^ base peak at *m*/*z* 377 was formed by losing a hexose residue [M − 162 − H]^−^ while the ion at *m*/*z* 307 by losing the C_4_H_6_O residue [M − 162 − 70 − H]^−^ from the released aglycone. Apart from oleuropein itself, two oleuropein isomers (**16**, **20**), both with a similar MS fragmentation pattern, were also found in the studied leaf extract. Based on the known elution order of oleuropein isomers [30], compound **20** was tentatively identified as oleuroside.

The analyzed extract also contained seven oleuropein derivatives (**5**, **8**, **9**, **15**, **17**, **18**, **21**) previously described in olive leaves [26,27,28,29]. The MS analysis of compound **5** revealed deprotonated ion [M − H]^−^ at *m*/*z* 377, and MS^3^ fragments at *m*/*z* 197 and 153, which allows the identification of oleuropein aglycone [26,27,29]. The compounds **8** and **17**, exhibiting different parent ions [M − H]^−^ at *m*/*z* 555 and 569, respectively, and the same MS^2^ fragmentation pattern with fragment ions at *m*/*z* 537, 403, and 223, were identified based on the literature [27,28] as hydroxyoleuropein (**8**), and methoxyoleuropein (**17**), respectively. The MS spectrum of compound **21** revealed the parent ion [M − H]^−^ at *m*/*z* 523 and MS^2^ fragments at *m*/*z* 361 (loss of a hexose residue), *m*/*z* 291 (loss of C_4_H_6_O) and *m*/*z* 259 (loss of C_4_H_6_O and CH_3_OH), typical for ligstroside [27,28,29]. Three other derivatives, **9**, **15** and **18**, were identified as isomeric oleuropein hexosides based on their deprotonated ions [M-H]^-^ at *m*/*z* 701 and MS^2^ fragments at *m*/*z* 539 and 377, formed by loss of hexose moieties and corresponding to oleuropein and its aglycone, respectively [27,28,29].

The several compounds previously described as hydrolysis products of oleuropein [23,24,25] were also present in the studied olive extract. The MS spectrum of compound **1** revealed the parent ion [M-H]^-^ at *m*/*z* 315 and the secondary ion at *m*/*z* 135, which is characteristic of hydroxytyrosol hexoside [27,28]. The compounds **2** and **3**, both with the [M − H]^−^ deprotonated ions at *m*/*z* 389 and MS^2^ fragment ions at *m*/*z* 227 [M − 162]^−^ (a loss of hexose residue), were identified as oleaoside isomers [27,28]. The compounds **4** and **6** revealed the parent ions [M − H]^−^ at *m*/*z* 403 and MS^2^ ions [M − 162 − 18 − H]^−^ at *m*/*z* 223, corresponding to dehydrated elenolic acid [27,28]. Thus, they were identified as elenolic acid hexoside isomers.

#### 2.1.2. Phenylethanoids

The MS spectra of compounds **12** and **13** exhibited identical parent ions [M − H]^−^ at *m*/*z* 623, which subsequently formed the MS^2^ and MS^3^ fragments characteristic for phenylethanoid glycosides [27,28,29]. The MS^2^ base ion [M − 162 − H]^−^ at *m*/*z* 461 and MS^3^ fragment ion [M − 162 − 146 − H]^−^ at *m*/*z* 315 were obtained by the loss of caffeoyl and rhamnose units. The MS^3^ fragment at *m*/*z* 135, formed by further loss of hexose and water units, corresponded to the 2-dihydroxyphenylethanol moiety [27,28,29]. Accordingly, the compounds were assumed to be verbascoside isomers, and finally confirmed by comparison with the authentic standard.

#### 2.1.3. Flavonoids

Among flavonoids, the third major group of phenolic compounds in *O. europaea* leaves [24,25], five compounds (**7**, **10**, **11**, **14**, **22**) were found in the analyzed extract, including four glycosides and one aglycone (**22**), which was identified with authentic standard as luteolin [24,27]. Glycosides were discriminated by the neutral losses of sugar moieties (-162 for hexose and -146 for rhamnose units) and by the signals at *m*/*z* 269 or *m*/*z* 285, assignable to the aglycone units of apigenin and luteolin, respectively. On that basis, compounds **11** and **14** were identified as hexosides of luteolin and apigenin [27,28,29], compound **7** as luteolin dihexoside [27,29], and compound **10** as luteolin rhamnoside-hexoside [28], respectively. Moreover, compound **11** was identified as luteolin 7-*O*-glucoside by direct comparison with the reference standard.

#### 2.1.4. Quantitative Analysis of Bioactive Compounds in *O. europaea* Leaf Extract

The LC-MS profile of bioactive compounds of OLE covered 22 identified constituents, but only four of them, i.e., oleuropein (**19**), oleuropein isomers (**16**, **20**) and luteolin-7-*O*-glucoside (**11**), could be considered as chief components of the studied extract. The total content of these compounds (Table 1), determined by the RP-HPLC-PDA, was 26.20 ± 0.02% dw, with prevailing oleuropein (22.20 ± 0.03%) and its isomers (2.60 ± 0.01%). The level of oleuropein was 16 times higher than that of the primary component from the flavone group, i.e., luteolin 7-*O*-glucoside (1.40 ± 0.02%). It is worth noting that the observed oleuropein level followed the producer’s declaration (not less than 20%) and is high in comparison with the literature data, which indicated the oleuropein content in commercially available olive leaf extracts in a wide range of 6.0–21.0% [31]. The variation in the oleuropein level was usually explained by the differences in the olive variety and the diversity of climate and geographical area of olive cultivation [31,32]. The harvesting time may also be influential since olive leaves were proven to reach their maximum phenolic content in the cold season around December [32]. In addition, the extraction process and problems with the stability of commercial products may also contribute to the oleuropein variability [31]. Therefore, the high content of oleuropein in the analyzed extract, guaranteed by the quality control, might explain its potent biological effects observed in the present study.

### 2.2. Prevention of UVA-Induced Death of Fibroblasts by Inhibition of the Intrinsic and Extrinsic Apoptosis Pathway

It is believed that the UVA photons are only partly absorbed by the upper layers of skin and can penetrate deep into the dermis by contrast with UVB radiation which is mostly absorbed by several major cutaneous targets in the epidermis [33]. Fibroblasts, like keratinocytes and melanocytes, are model cells used in many in vitro studies on photoprotection mechanisms. In our study we used cultured fibroblasts, which are often used to assess the effects of UVA radiation and the photoprotective action of plant compounds. The study conducted by Battie et al. suggested that dermal fibroblasts are more sensitive to UVA-induced oxidative stress than keratinocytes [34]. Fibroblasts are the primary dermal cells responsible for producing extracellular matrix components determining the structural and mechanical properties of the skin. Many studies have demonstrated that fibroblasts in a monolayer culture reproduce various forms of UV-induced photodamage at the cellular level [35]. To prevent damage caused by UVA-radiation much attention has focused on plant extracts and compounds with antioxidant properties. Here, we have demonstrated that pretreatment of fibroblasts with OLE efficiently ameliorated the UVA-induced toxic effects. Quercetin (QU) and ascorbic acid (AA) are among the most powerful polyphenolic and water-soluble antioxidants, respectively. These antioxidants served as the reference plant compounds in the study. It has been well established that vitamin C limits the damage induced by UV exposure [36]. This type of injury is directly mediated by a radical-generating process, and protection is primarily related to its antioxidant activity. This has been demonstrated in vitro and in vivo, using both topical and dietary intake of vitamin C. Quercetin demonstrated protection against UV-induced oxidative damage, and it modulated cell signal transduction pathways, but the efficacy of quercetin is limited by its water-insolubility and therefore its poor penetration and deposition in the skin [3].

A possible adverse effect of OLE on viability of human dermal fibroblasts (Hs68) was assessed using the CCK-8 assay in a concentration range of (2.5–25 μg/mL) after 24 h treatment. As shown in Figure 2A, there was no significant reduction in cellular viability, which imply that OLE at doses applied in experiments was not cytotoxic. Next, Hs68 fibroblasts were incubated with different concentrations of OLE (2.5–25 μg/mL, 24 h) prior to UVA irradiation (8 J/cm^2^), and cell viability was analyzed by CCK-8 assay. It was found that UVA irradiation significantly decreased the viability of dermal fibroblasts, the cell survival fraction was approximately 80% (*p* < 0.05). However, cells subjected to OLE pretreatment exhibited sufficient pro-survival against UVA phototoxicity (OLE entirely reversed the decline of cell viability (Figure 2B). The optimal intensity of UVA was experimentally determined by treating cells with increasing doses of UVA (2, 4 and 8 J/cm^2^). Since cell viability was not affected with doses 2 and 4 J/cm^2^ the dose of 8 J/cm^2^ was finally selected. To be effective, the concentration of the photoprotective agent strongly depends on the UV radiation dose used in the experiment. Therefore, the concentration range of OLE was experimentally established at the beginning of the study based on the determination of the level of DNA damage and the viability of fibroblasts after UVA-irradiation (using the CCK-8 method), and then consistently used in further study. The cell viability results were confirmed in annexin V/propidium iodide staining experiments followed by the flow cytometry analysis, where the comparable cytotoxic effect of UVA was revealed (Figure 2C). It must be underlined that under our experimental conditions the irradiation of fibroblasts with a dose of UVA 8 J/cm^2^ caused a significant increase in the percentage of overall apoptosis (either cells positive stained with annexin V or both annexin V and PI positive stained cells) with negligible fraction of necrosis (PI-positive stained cells) (Figure 2D).

Apoptosis is the best characterized form of programmed cell death due to its importance in development, homeostasis, and in pathogenesis of many diseases, including photoaging and skin cancers. Cells respond to specific apoptotic signals by initiating intracellular processes that result in a series of morphological and biochemical changes such as membrane blebbing, cell-shrinkage, loss of mitochondrial membrane potential, chromatin condensation and DNA fragmentation. One of the early apoptotic events is the externalization of phosphatidylserine (PS) on the outer leaflet of the plasma membrane. Annexin V, a Ca^2+^-dependent phospholipid-binding protein with high affinity for PS, binds to exposed apoptotic cell surface PS. Therefore, Annexin V conjugated with fluorescein isothiocyanate (FITC) serves as a sensitive probe for flow cytometric analysis of undergoing apoptosis in cells. The Hs68 cells were pre-treated with OLE before UVA exposure, and the assay was performed after 24 h post-radiation. The results were expressed as the apoptotic index, and to calculate of which the number of apoptotic cells they were divided by the number of cells with no measurable apoptosis. As shown in Table 2, OLE at concentrations of 5 and 25 μg/mL largely prevented fibroblast apoptotic death (by above 50% compared to control, *p* < 0.001).

To further investigate the molecular mechanism responsible for the inhibitory effect of OLE on UVA-induced apoptosis in Hs68 cells, the activities of caspase-3, -8 and -9 were evaluated by the colorimetric assay kits using specific substrates. Our results show that the activities of caspase-8 and -9 were elevated approximately 6-fold (the absorbance at 450 nm, raised from 0.049 ± 0.007 to 0.200 ± 0.010 and from 0.065 ± 0.015 to 0.385 ± 0.047, respectively) and of caspase-3 about 4.5-fold (A_450_ increased from 0.054 ± 0.006 to 0.241 ± 0.010) in fibroblasts irradiated with a dose of 8 J/cm^2^, compared to untreated cells. These results suggest that both the intrinsic and extrinsic apoptotic pathways appear to be activated after UVA stimulation. Moreover, the data demonstrate that in fibroblasts pre-incubated with OLE the activities of all three caspases were significantly inhibited. The inhibitory effect of OLE was concentration-dependent. Interestingly, at the high concentration used (25 μg/mL) the caspase-9 activity was reduced by as much as 80% (*p* < 0.001) (Figure 3C), and caspase-3 and -8 activities were inhibited by above 50% (*p* < 0.001) (Figure 3A,B). The reference antioxidants, quercetin and ascorbic acid, displayed the comparable effects. Our results remain in line with the earlier findings of Salucci et al. [21] who have shown the activation of intrinsic and extrinsic apoptotic pathways in UVB-irradiated human keratinocytes, and their down-regulation when antioxidants (i.e., hydroxytyrosol) were added to cells before death induction. The intrinsic apoptotic signaling pathway can be induced by the excessive production of ROS and is responsible for opening pores in the mitochondrial membrane leading to cytochrome c release [37]. Cytochrome c with cytoplasmic APAF-1 participates in the activation of caspase-9. Since caspase-9 activity was found to be most inhibited it is reasonable to hypothesize that OLE can decrease fibroblast apoptosis rate via the attenuation of the DNA damage and inhibition ROS generation, the two main factors responsible for apoptosis trigger.

### 2.3. Suppression of the DNA Damage and ROS Generation in UVA-Irradiated Fibroblasts

To clarify whether OLE can protect Hs68 cells against the UVA-induced genotoxicity, the degree of DNA damage was monitored by using the alkaline version of the comet assay. A variety of DNA lesion, i.e., single- and double-strand breaks (SSBs and DSBs) or alkali-labile sites, contribute to the total DNA damage that can be detected by the alkaline comet assay. DNA damage, mostly DSBs, are thought the most cytotoxic lesion induced by radiation and, if not properly repaired, are known to trigger cell death. Although, cell exposure to UVA radiation generates mostly thymine cyclobutane dimers [2], which are formed via a direct photochemical mechanism without mediation of a cellular photosensitizer, it has been reported that UVA-induced DSBs can be generated from the repair of clustered oxidative DNA damages [24]. As shown in Figure 4, UVA irradiation (8 J/cm^2^) induced a significant increase in the total DNA damage (measured as a % tail DNA). However, pre-treatment of fibroblasts with OLE resulted in a concentration-dependent reduction in the percentage of tail DNA. At the optimal extract concentration of 25 µg/mL, the level of DNA damage decreased up to less than 5%, which was similar to that of QU and AA; *p* < 0.001).

UVA radiation induces excessive ROS generation such as singlet oxygen (^1^O_2_), superoxide anion (O_2_**^.^**^−^), hydrogen peroxide (H_2_O_2_), singlet oxygen (^1^O_2_) and hydroxyl radicals (OH), which is the principle cause of oxidative damage of vital molecules (including DNA, protein and membrane lipids) in irradiated cells [3]. To verify if the antioxidant capacity of OLE contributes to protection against UVA-induced cell death, we assessed the intracellular ROS production. The ability of OLE to reduce ROS generation induced by UVA in Hs68 cells was evaluated using H_2_DCF-DA (Figure 5). H_2_DCF-DA is a nonfluorescent probe that is deesterified by intracellular esterases and is oxidized by intracellular ROS to fluorescent 2′,7′-dichlorofluorescein [18]. The amounts of intracellular ROS were evaluated by the changes in DCF fluorescence intensity, the effects of OLE on ROS generation were expressed as a percent of control, where fluorescence intensity in UVA-irradiated fibroblasts (untreated with the extract) was taken as 100%. In our experimental conditions, UVA increased ROS production by above 4-fold (the increase in fluorescence intensity from 0.556 ± 0.138 to 2.296 ± 0.317 has been noted). Pretreatment with 5 and 25 μg/mL OLE reduced ROS production by near 30% compared to irradiated and nontreated cells. To the best of our knowledge, the antioxidant activity of OLE in UVA-irradiated fibroblasts has only been investigated by Czerwinska et al. [38] who established, using the same assay as in our study (H_2_DCF-DA), that the aqueous and ethanolic extracts from *O. europaea* leaves did not show any significant inhibition of ROS production. The reason for these discrepancies can most likely be some differences in the composition of the extracts and/or the stability of their constituents in the presence of UV radiation. On the other hand, the authors’ study showed that the aqueous extract at the concentration of 5 and 25 μg/mL did have a protective effect on the viability of UVA-treated fibroblasts.

### 2.4. Enhancement of the Intracellular Thioredoxin Reductase in UVA-Irradiated Fibroblasts

Excessive ROS production invokes a plethora of defense mechanisms controlling the redox status to protect against oxidative damage. These include small radical trapping molecules, i.e., vitamins A, C, E, thiols and antioxidant enzymes. It has been suggested that thioredoxin/thioredoxin reductase (Trx/TrxR) system provided one of the skin’s first defenses against ROS generated in response to UV light [39]. Thioredoxin is a cytokine-like factor mainly located in the cytoplasm that quickly translocates into the nucleus in response to oxidative stress. Moreover, the presence of several thioredoxin reductases has been demonstrated in human skin, which are the members of the family of homodimeric proteins where each monomer includes an FAD prosthetic group and an NADPH binding site. TrxR reduces hydrogen peroxide, lipid peroxides and quinones [39]. TrxR is one of many transcriptional targets of the tumor suppressor gene p53 [39].

We investigated the influence of OLE on the level of TrxR1 (the cytosolic TrxR form) in fibroblasts. First, we showed that the extract alone slightly reduced TrxR expression (*p* < 0.001) in dermal fibroblasts (Figure 6). UVA-radiation significantly reduced the level of TrxR (by approximately 45%, compared to control; *p* < 0.001). The TrxR levels in UVA-irradiated fibroblasts were significantly elevated by OLE, as well as by QU treatments, which suggest a vital role of this enzyme in the cytoprotective activity of the plant extract (like QU). This finding remains consistent with the recent study conducted by Huang et al. [40] who have shown significantly reduced Trx and TrxR activities in human keratinocytes after UVB radiation. In addition, their results revealed that the bamboo extract, from *Acidososa longiligula*, alleviated UV-induced apoptosis and ROS production via the positive regulation of Trx1.

### 2.5. Anti-Inflammatory Activity of OLE

Pro-inflammatory cytokines such as TNF-α and interleukin 2 (IL-2) play an important role in the acute and chronic inflammatory response, respectively [41]. A previous study found a significant decrease in TNF-α secretion level from PBMCs upon OLE treatment at concentration of 80 μg/mL after stimulation with lipopolysaccharide [13], in addition, oleuropein was identified as the only OLE component responsible for this anti-inflammatory effect. Therefore, we aimed to examine the effects of OLE on TNF-α and IL-2 secretion levels in UVA-irradiated PBMCs. First, we used concanavalin A to stimulate PBMCs. Con A is an antigen-independent mitogen frequently used as a surrogate for antigen-presenting cells in T cell stimulation experiments. We observed that UVA irradiation (8 J/cm^2^) clearly increased the level of TNF-α, and this stimulating effect was comparable with the con A action (Figure 7A). OLE, similar to indomethacin, the reference potent nonsteroidal anti-inflammatory drug, significantly inhibited (by more than 90%, *p* < 0.001) the con A-stimulated secretion of TNF-α. Further experiments showed that the secretion of both cytokines (TNF-α and IL-2) were significantly reduced in UVA-stimulated PBMCs by the treatment with OLE at 25 μg/mL and 5/25 μg/mL (*p* < 0.001), respectively (Figure 7B,C). However, the noticeable inhibitory effect was not as strong as in the case of stimulation with con A, which can result from different mechanisms responsible for cytokine secretion. Nevertheless, these data prove the anti-inflammatory activity of OLE.

## 3. Materials and Methods

### 3.1. Reagents

l-ascorbic acid, bovine serum albumin (BSA), Cell Counting Kit-8 (CCK-8), concanavalin A, DAPI (4′,6-diamidino-2-phenylindole dihydrochloride), 2′,7′-dichlorofluorescin diacetate (H_2_DCF-DA), dimethyl sulfoxide (DMSO), Histopaque-1077, hydrogen peroxide, low melting-point (LMP) agarose, normal melting-point (NMP) agarose, penicillin-streptomycin solution, quercetin (≥95%), and RPMI-1640 medium without glutamine were purchased from Sigma-Aldrich Chemicals (St. Louis, MO, USA). Dulbecco’s modified Eagle medium (DMEM) with 4.5 g/L glucose and l-glutamine was obtained from Lonza (Basel, Switzerland). Dulbecco’s Phosphate Buffered Saline (DPBS) without calcium and magnesium, Hanks BSS (HBSS) without phenol red with calcium and magnesium, and Phosphate Buffered Saline (PBS) were purchased from Biological Industries (Cromwell, CT, USA). Heat-inactivated fetal bovine serum (FBS) and trypsin-EDTA solution were from Biowest (Nuaillé, France). Trypan blue was obtained from LifeTechnologies (Waltham, MA, USA). All other analytical grade and high-quality chemicals were obtained from local commercial suppliers, such as Chempur (Piekary Slaskie, Poland) or POCH S.A. (Gliwice, Poland).

This study was approved by the local Ethics Committee (no. 12/KBBN-UŁ/III/2018).

### 3.2. Plant Extract

The commercial food supplement *Olive leaf*, containing the dry extract obtained from the leaves of *Olea europaea* (Oleaceae), was purchased from the local herbal company Medica Herbs (Krakow, Poland). The commercial preparation comprises 60 capsules, each with 520 mg of ethanol-water (2:3, *v*/*v*) extract of *O. europaea* leaves, standardized to 20% oleuropein content (following the producer’s declaration).

### 3.3. Phytochemical Profiling

The qualitative UHPLC-PDA-ESI-MS^3^ analysis was performed on the UHPLC-3000 RS system (Dionex, Dreieich, Germany) equipped with a dual low-pressure gradient pump, a diode array detector, and an AmaZon SL ion trap mass spectrometer with an ESI interface (Bruker Daltonics, Bremen, Germany). Separations were carried out on a Kinetex XB-C18 column (150 × 2.1 mm, 1.7 µm; Phenomenex Inc., Torrance, CA, USA) at 25 °C with a flow rate of 0.3 mL/min. The mobile phase consisted of solvent A (water/formic acid, 100:0.1, *v*/*v*), and solvent B (acetonitrile/formic acid, 100:0.1, *v*/*v*) with the following elution profile: 0–45 min, 6–26% B; 45–55 min, 26–95% B; 55–60 min, 95% B; 60–63 min 95–6% B. The samples were dissolved in methanol-water (7:3, *v*/*v*) to the final concentration of the extract solution 5 mg/mL. The LC eluate was introduced directly into the ESI interface without splitting and analyzed in a negative ion mode using a scan from *m*/*z* 70 to 2200. The nebulizer pressure, the dry gas flow, the temperature, and the capillary voltage were 40 psi, 9 L/min, 300 °C, and 4.5 kV, respectively. The MS/MS fragmentation amplitude was set to 1.0 V and automatically adjusted by the system in the range of 60–300% of this value.

The quantitative HPLC-PDA analysis was performed on the Waters 600E Multisolvent Delivery System (Waters, Milford, MA, USA) with a PDA detector (Waters 2998) scanning in the wavelength range of 220–550 nm. Separations were carried out on a C18 Ascentis^®^ Express column (75 × 4.6 mm, 2.7 µm; Supelco, Bellefonte, PA, USA), guarded by a C18 Ascentis^®^ C18 Supelguard column (20 × 4 mm, 3 µm; Supelco), at 25 °C with flow rate 1.4 mL/min. The mobile phase consisted of solvent A (water/orthophosphoric acid, 99.5:0.5 *v*/*w*) and solvent B (acetonitrile) with the elution profile as follows: 0–16 min, 15–30% B; 16–17 min, 30–50% B; 17–19 min, 50% B; 19–20 min, 50–15% B; 20–23 min, 15% B (equilibration). All gradients were linear. The samples were dissolved in methanol-water (7:3, *v*/*v*) to the final concentration of the extract solution 1.3 mg/mL. The dominant components of the extracts were quantified using external standards of oleuropein (λ = 280 nm, linearity range 40.1–401.0 µg/mL, R^2^ = 0.9996) and luteolin 7-*O*-glucoside (λ = 340 nm, linearity range 6.0–60.0 µg/mL, R^2^ = 0.9996).

### 3.4. Cell Culture

Human skin (foreskin) fibroblast cell line (Hs68; ATCC^®^ CRL-1635™) was purchased from the American Type Culture Collection (ATCC; Manassas, VA, USA). Cells were grown in DMEM supplemented with 10% heat-inactivated FBS and 1% penicillin-streptomycin solution (10,000 units penicillin and 10 mg streptomycin/mL) in a humidified (90–95%) atmosphere with 5% CO_2_, at 37 °C. Cells were cultured until they formed a confluent monolayer and were harvested by trypsinization using the trypsin-EDTA solution (0.25% trypsin).

### 3.5. Cell Treatment and UVA Irradiation

The stock solution of the olive leaf extract (OLE) (20 mg/mL) was prepared in DMSO and then diluted with DMEM to obtain the concentrations 20-fold higher than required (50–500 μg/mL). For each experiment, Hs68 cells were seeded onto 96- or 6-well (for the comet assay) culture microplates in DMEM and were grown for 24 h. For treatments, the medium was removed and the fresh serum-free DMEM was added (95 μL), supplemented with 5 μL of OLE (to obtain the final concentrations 2.5–25 μg/mL), the reference antioxidants (final concentrations 25 μg/mL) such as ascorbic acid (AA) and quercetin (QU) or with vehicle (0.125% DMSO), and fibroblasts were kept in the incubator (5% CO_2_, at 37 °C) for 24 h.

Prior to UVA irradiation, cells were washed twice with DPBS and exposed to UVA irradiation in a thin layer of DPBS (opened covers). Fibroblasts (in microplates placed on ice) were irradiated at a dose of 8 J/cm^2^. The UVA radiation source was a low-pressure mercury lamp (Emita VP-60, Famed, Lodz, Poland) equipped with a filter transmitting the radiation of the UVA fraction (emission peak at 365 nm). Immediately after irradiation, Hs68 cells were fed with a fresh serum-free DMEM and allowed to grow under standard culture conditions for another 24 h. At the same time (in all experiments) a duplicate set of Hs68 cell samples was prepared, which was subjected to the same sham-irradiation conditions.

### 3.6. Cell Viability

Cell viability was assessed using the colorimetric assay, cell counting kit-8 (CCK-8), performed according to the manufacturer’s instructions. A water-soluble tetrazolium salt (WST-8) to produce a water-soluble formazan dye upon reduction in the presence of an electron carrier is used in the assay. The amount of a formazan dye generated by the activity of dehydrogenases in cells is directly proportional to the number of living cells. Hs68 cells (5000 per well) grown in 96-well plates for 24 h were exposed to OLE followed by UVA-irradiation under conditions described before. After a 24-h culture, cells were washed twice with DPBS, then 100 μL of fresh media and CCK-8 solution (10 μL) were added to each well. After the incubation for 4 h (at 37 °C, 5% CO_2_) the absorbance at 450 nm was determined in the microplate reader SPECTROstar^®^ Nano (BMG LABTECH GmbH, Ortenberg, Germany).

### 3.7. Comet Assay

To evaluate the genotoxic effect of UVA light on Hs68 cells, the comet assay under alkaline conditions was performed according to the procedure of Singh et al. [42] with some modifications [43]. Hs68 cells, seeded onto 6-well plate (2 × 10^5^ cells per well) after 24-h incubation with OLE followed by the exposure to UVA irradiation (as described above) were scraped off the bottom of a well with a cell scraper and the cell suspension was centrifuged (180× *g*, 15 min, ambient temperature). A pellet of cells was suspended in 1.0% LMP agarose and spread onto microscope slides pre-coated with 0.5% NMP agarose. Next, the cells were lysed for 1 h, at 4 °C in a lysis buffer (2.5 M NaCl, 100 mM EDTA, 1% Triton X-100, 10 mM Tris, pH 10). All remaining steps including electrophoresis, DAPI comet staining, and measuring were performed as described previously [44]. The number of comets was counted on a Nikon fluorescence microscope (Nikon Instruments Inc., Melville, NY, USA) connected to a Cohu 4910 Series High-Performance Monochrome CCD Camera (Cohu Inc., Montreal, QC, Canada) using the Lucia-Comet v. 4.51 software. Cells suspended in a fresh DMEM-containing vehicle (0.125% DMSO) served as a negative control. A positive control was prepared by treating cells with 10 µM hydrogen peroxide.

### 3.8. ROS Level

The intracellular ROS generation induced by UVA radiation was spectrofluometrically monitored using the H_2_DCF-DA probe [45]. Briefly, Hs68 cells (1 × 10^4^ cells per well) after treatment with OLE (24 h) were incubated with 10 µM DCFH2-DA for 30 min (37 °C, 5% CO_2_) in 96-well black with clear bottom culture microplates (Thermo Scientific, Waltham, MA, USA). Next, the cells were washed twice with Hank’s Balanced Salt Sodium buffer (HBSS, Biowest, Nuaillé, France), resuspended in HBSS (50 µL), and subjected to UV-A radiation (8 J/cm^2^). Fibroblasts treated with 10 µM H_2_O_2_ were used as positive controls, cells suspended in HBSS (untreated) as negative controls. The fluorescence intensity was read at λex = 485 nm excitation and λem = 538 nm emission, after a 120-min incubation (37 °C) using a microplate fluorometer FluoroskanTM FL (ThermoFisher Scientific, Waltham, MA, USA).

### 3.9. Thioredoxin Reductase Level

Thioredoxin reductase (TrxR) concentration was determined using the commercial Thioredoxin Reductase 1 (TXNRD1) Human SimpleStep ELISA^®^ Kit (ab192150; Abcam, Cambridge, UK) according to the manufacturer’s protocol. In brief, Hs68 cells (1 × 10^6^ cells per well) were treated as above by OLE followed by UVA. The cells were homogenized in assay buffer, centrifuged (10,000× *g*, 15 min, 4 °C) and the supernatant was collected. The protein concentration of the supernatant was determined using the Bradford Reagent (Sigma-Aldrich Chemicals, St. Louis, MO, USA). The absorbance at 450 nm was read in the microplate reader SPECTROstar^®^ Nano (BMG LABTECH GmbH, Ortenberg, Germany).

### 3.10. Annexin V (FITC) Apoptosis

Apoptosis was measured using the Annexin V-fluorescein isothiocyanate (FITC) Apoptosis Detection Kit (Biovision, Milpitas, CA, USA) according to the instructions of the manufacturer. Briefly, 1 × 10^5^ cells, resuspended in 500 μL of 1× Binding Buffer, were mixed with 5 μL of FITC conjugated annexin V and 5 μL of propidium iodide (PI). After the 15 min incubation in the dark, the stained cells were analyzed by flow cytometry. The intensity of fluorescence was measured in 30,000 cells, using the flow cytometer CyFlow Cube 6 (Sysmex Partec GmbH, Görlitz, Germany), with the laser excitation wavelength of 488 nm and a 530 nm emission filter. Data were analyzed with the Windows™-based FCM software CyFlow.

### 3.11. Measurement of Caspase-3, -8 and -9 Activities

Caspase-3, caspase-8 and caspase-9 enzymatic activities in Hs68 cells were determined using the commercial Caspase-3/CPP32, FLICE/Caspase-8 and Caspase-9 Colorimetric Assay Kits, respectively (BioVision, Mountain View, CA, USA). The assays were performed according to the manufacturer’s instructions. Cells were seeded (1 × 10^6^ cells/well) onto 96-well culture microplates, and then treated as above by OLE followed by UVA. The culture medium served as the blank, the cytosolic extract of untreated cells as the negative control. Substrate cleavage, which resulted in the release of pNA (405 nm), was measured in the microplate reader SPECTROstar^®^ Nano (BMG LABTECH GmbH, Ortenberg, Germany). Comparing the absorbance of pNA from the apoptotic sample with the non-induced control allowed the degree of increase in caspase activity to be determined.

### 3.12. Pro-Inflammatory Cytokine Production in PBMCs

Human peripheral blood mononuclear cells (PBMCs) were isolated from buffy coats, purchased at Regional Center for Transfusion Medicine in Lodz (Poland), by centrifugation in a density gradient of Histopaque-1077 as described earlier [46]. PBMCs were suspended in RPMI 1640 medium, supplemented with 10% fetal calf serum and 0.1% of penicillin-streptomycin, at the density of 1.5 × 10^6^ cells/mL and cells were pre-incubated for 1h with OLE (or vehicle) in a laboratory CO_2_ incubator (in 96-well microplates, at 37 °C, with 5% of CO_2_ concentration). Then, PBMCs were either stimulated with concanavalin A (Con A; 10 μg/mL final concentration) or UVA-irradiated (8 J/cm^2^) and cultured for 24 h. Next, microplates were centrifuged (540× *g*, 15 min) and the supernatants were collected for further analyses.

The levels of tumor necrosis factor α (TNF-α) and interleukin-2 (IL-2) secreted to the PBMC medium were detected using the commercial ELISA kits (Human TNF-α GENLISA KB1145; Human IL-2 GENLISA KB1064, KRISHGEN BioSystem, Athens, GA, USA, respectively) according to protocols provided by the manufacturer. Assays were carried out in triplicate, using indomethacin (25 μg/mL) as a reference.

### 3.13. Statistical Analysis

All obtained results are presented as mean values ± SD. Using Shapiro–Wilk test, the normality of the results was examined. Then, the non-parametric Levene’s test for homogeneity of variance was performed. Based on the Levene’s test (ANOVA) followed by post-hoc Tukey test the differences between values were evaluated. All obtained data were analysed using StatSoft Inc. “Statistica” v. 13.1 (TIBCO Software Inc. Palo Alto, CA, USA). The value of *p* < 0.05 are statistically significant. All presented figures were prepared using GraphPad Prism 5 Software v.5.01 (Dotmatics, San Diego, CA, USA).

## 4. Conclusions

It has been well established that exposure of skin to UV radiation results in inflammation, DNA damage, oxidative stress, and activation of cell signaling pathways leading to photodamage and/or development of skin cancer. The results obtained in the present work show that OLE, a commercial food supplement standardized to 20% oleuropein content, efficiently protected human skin fibroblasts against the UVA-induced oxidative stress and DNA damage and was able to increase the post-radiation viability of fibroblasts by inhibiting their apoptotic death. We hypothesized that the TrxR up-regulation by OLE could have prevented the UVA-induced apoptotic fibroblast death.

Our in vitro study proved OLE to be a potential candidate for the protection of fibroblasts against the damaging effects of UVA irradiation. In further studies an assessment of the in vivo protective efficacy of OLE is necessary.

## Figures and Tables

**Figure 1 molecules-27-05144-f001:**
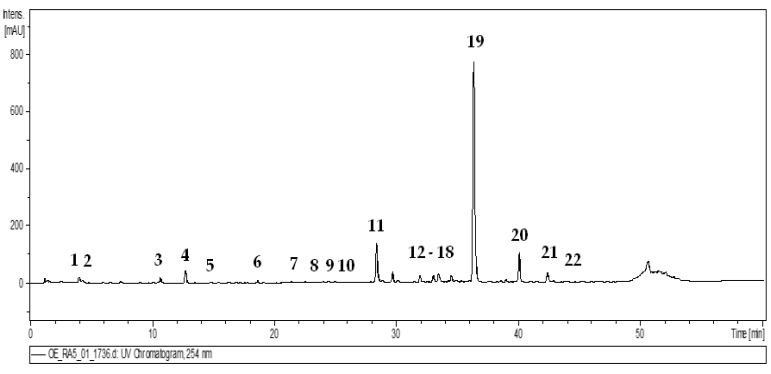
Representative UHPLC-UV chromatogram of the *O. europaea* leaf extract at 254 nm. The peak numbers refer to those presented in Table 1.

**Figure 2 molecules-27-05144-f002:**
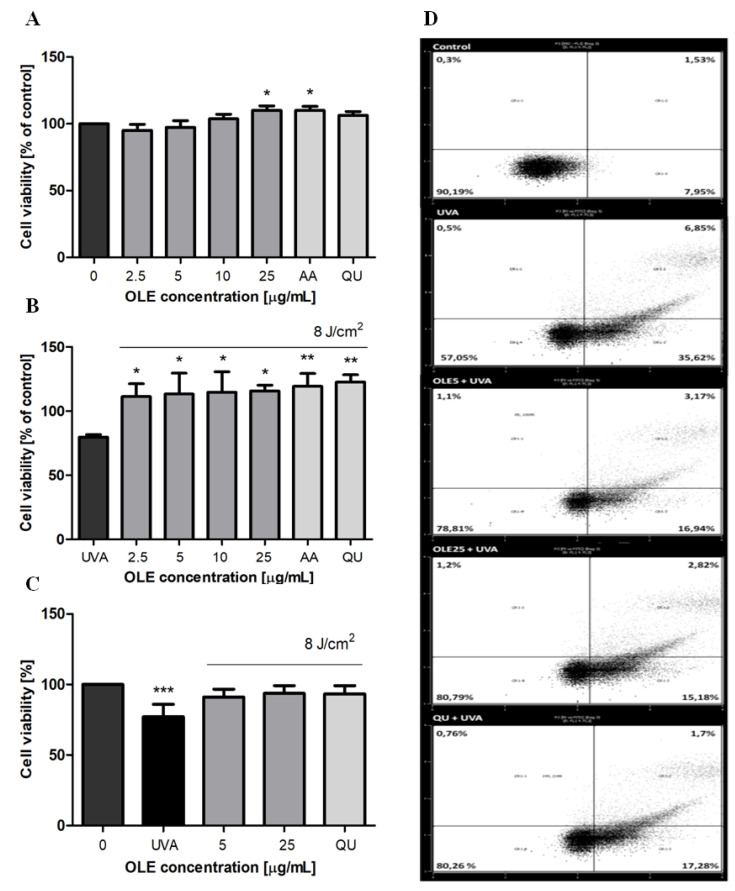
The effects of different concentrations of OLE on the viability of human skin fibroblasts. Cells cultured (24 h) in the presence of OLE (2.5–25 μg/mL), vehicle (0), or the reference antioxidants AA and QU (25 μg/mL) not irradiated (**A**) or exposed to UVA-irradiation (8 J/cm^2^) (**B**) were cultured for an additional 24 h, and then subjected to CCK-8 assay. The cytotoxicity of OLE and fibroblast survival were expressed as a percentage of control, where the absorbance of untreated and of corresponding non-irradiated cells were taken as 100%, respectively. The post-radiation viability of fibroblasts evaluated by the Annexin V-FITC/PI staining (flow cytometry) assay (**C**). A representative dot-plot histogram is shown (**D**), cell populations in II, I and III/IV quadrants represent alive, undergoing necrosis and undergoing apoptosis Hs68 cells, respectively. The figure shows mean results from 3–6 independent experiments. Error bars denote ±SD. * *p* < 0.05; ** *p* < 0.01; *** *p* < 0.001 compared to control (0).

**Figure 3 molecules-27-05144-f003:**
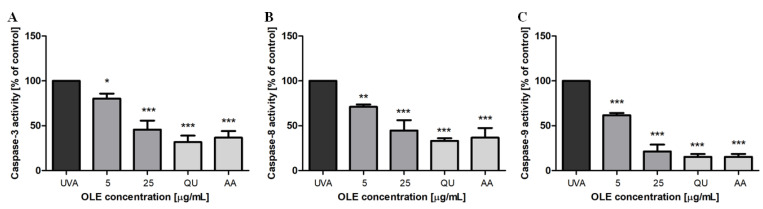
Inhibitory effect of OLE on UVA-induced activation of caspase-3 (**A**), -8 (**B**) and -9 (**C**) in human skin fibroblasts. Caspase activities are expressed as % of control, where the caspase activity in untreated and UVA-irradiated cells were taken as 100%. The figure shows mean results from 3 independent experiments. Error bars denote ±SD. * *p* < 0.05, ** *p* < 0.01, *** *p* < 0.001 compared to cells irradiated in the absence of OLE or the reference antioxidants (UVA).

**Figure 4 molecules-27-05144-f004:**
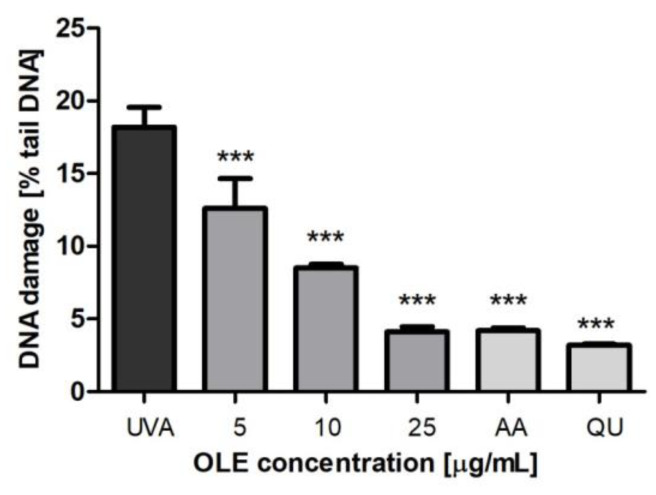
DNA damage, measured as the mean comet tail DNA (%) of human skin fibroblasts, untreated (UVA) or pre-incubated (24 h) with OLE (5–25 μg/mL) or the reference antioxidants AA and QU (25 μg/mL) followed by exposure to UVA (8 J/cm^2^). The values of comet tail DNA were reduced by the values obtained in comet assay for non-irradiated Hs68 cells (without or with the appropriate plant compounds); the extent of this endogenous DNA damage for control (not displayed in the figure), expressed as a percentage of tail DNA, was 1.61 ± 0.32%. The number of cells in each treatment was 100. The figure shows mean results from 3 independent experiments. Error bars denote ±SD. *** *p* < 0.001 compared to cells irradiated in the absence of OLE or antioxidants (UVA).

**Figure 5 molecules-27-05144-f005:**
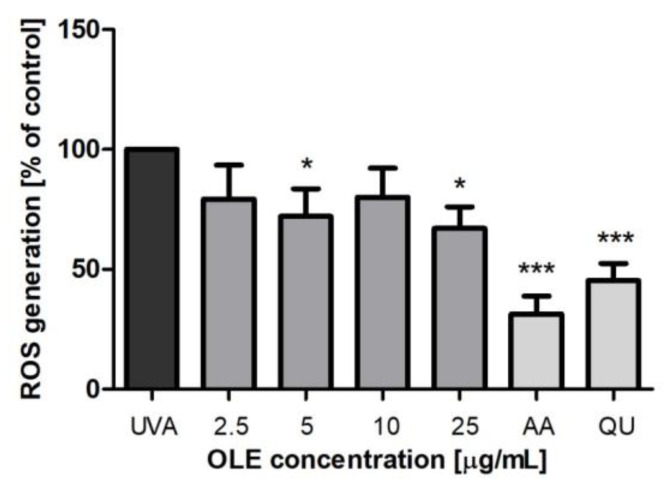
Changes of the intracellular ROS production in human skin fibroblasts, untreated (control) or pre-incubated (24 h) with OLE (2.5–25 μg/mL) or reference antioxidants AA and QU (25 μg/mL) followed by exposure to UVA (8 J/cm^2^). The figure shows mean results from 5 independent experiments. Error bars denote ± SD. * *p* < 0.05; *** *p* < 0.001 compared to cells irradiated in the absence of OLE (UVA).

**Figure 6 molecules-27-05144-f006:**
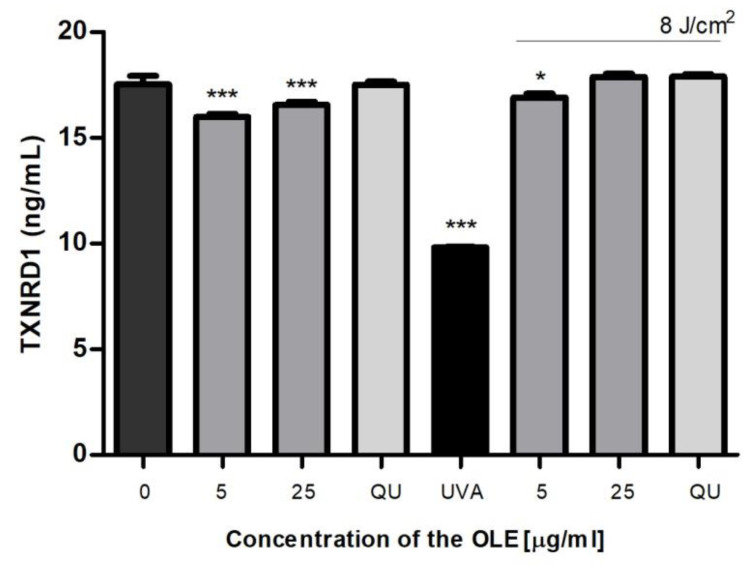
The level of thioredoxin reductase (TxnR) in human skin fibroblasts untreated (0) or pre-incubated (24 h) with OLE (5 and 25 μg/mL) or reference antioxidants (25 μg/mL) and exposed (or not exposed) to UVA (8 J/cm^2^). Mean values ± SD of three independent experiments is presented. Statistically significant differences vs. non-treated group (control); * *p* < 0.05; *** *p* < 0.001.

**Figure 7 molecules-27-05144-f007:**
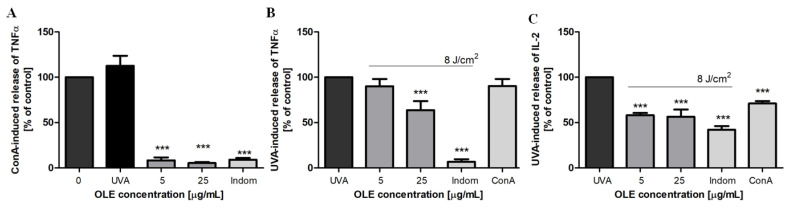
The effects of OLE on the concanavalin A (**A**) or UVA-induced (**B**) secretion of TNF-α and UVA-induced IL-2 release (**C**) by human peripheral blood polymorphonuclear cells. The results are expressed as % of control, where the cytokine level in untreated stimulated with ConA (10 μg/mL) or UVA (8 J/cm^2^) cells were taken as 100%. The figure shows mean results from 3 independent experiments. Error bars denote ±SD. *** *p* < 0.001 compared to stimulated cells in the absence of OLE (UVA). Indomethacin (25 μg/mL) was used as a reference.

**Table 1 molecules-27-05144-t001:** UHPLC-PDA-ESI-MS^3^ data of polyphenols identified in the commercial *O. europaea* leaf extract.

Peak	Analyte	t_R_ (min)	UV λ (nm)	Molecular Formula	[M − H]^−^ *m*/*z*	MS^2^ **	MS^3^	Lit.
	*Secoiridoids*							
**1**	Hydroxytyrosol hexoside	3.6	280	C_14_H_20_O_8_	315	153(1.4), 123(1.2), 95(0.2)		[27,28]
**2**	Oleaoside	4.2	235	C_16_H_22_O_11_	389	345(20), 227(100), 183(74), 165(60)		[27,28]
**3**	Oleaoside isomer	10.7	235	C_16_H_22_O_11_	389	345(100), 227(3.0), 209(32), 165(12)		[27,28]
**4**	Elenolic acid hexoside	12.7	245, 280	C_17_H_24_O_11_	403	371(21), 223(100), 179(40), 121(4.0)		[27,28]
**5**	Oleuropein aglycone	15.0	235, 280	C_19_H_22_O_8_	377	197(100), 153(11)		[26,27,29]
**6**	Elenolic acid hexoside isomer	18.8	245, 280	C_17_H_24_O_11_	403	371(100), 223(56), 179(37), 121(3.4)		[27,28]
**8**	Hydroksyoleuropein	24.5	235, 280	C_23_H_31_O_14_	555	537(100), **403**(50), 223(17)	371(25), 223(100), 179(52)	[27]
**9**	Oleuropein hexoside	27.5	235, 280	C_31_H_42_O_18_	701	539(32), 469(40), 437(21), 315(100)		[27,28,29]
**15**	Oleuropein hexoside	33.5	235, 280	C_31_H_42_O_18_	701	**593**(100), 377(14), 307(12), 275(12)	377(40), 307(100), 275(76)	[27,28,29]
**16**	Oleuropein isomer	34.4	235, 280	C_25_H_32_O_13_	539	403(37), 223(100), 179(32)		[26,27,28,29]
**17**	Methoxyoleuropein	35.0	235, 280	C_26_H_34_O_14_	569	537(100), 403(88), 223(13)		[27,28]
**18**	Oleuropein hexoside	35.9	235, 280	C_31_H_42_O_18_	701	539(100), 377(11), 307(10)		[27,28,29]
**19**	Oleuropein *	36.2	235, 280	C_25_H_32_O_13_	539	377(100), 345(11), 307(84), 275(74)		[26,27,28,29]
**20**	Oleuropein isomer	40.0	235, 280	C_25_H_32_O_13_	539	377(44), 345(11), 307(100), 275(99)		[26,27,28,29]
**21**	Ligstroside	42.3	230, 280	C_25_H_32_O_12_	523	361(100), 291(40), 259(24)		[27,28,29]
	*Phenylethanoids*							
**12**	Verbascoside *	29.7	235, 329	C_29_H_36_O_15_	623	**461**(100)	315(100), 135(36)	[27,28,29]
**13**	Verbascoside isomer	31.9	235, 329	C_29_H_36_O_15_	623	**461**(100)	315(100), 135(37)	[27,28,29]
	*Flavonoids*							
**7**	Luteolin dihexoside	21.7	340	C_27_H_30_O_16_	609	**447**(100)	285(100)	[28]
**10**	Luteolin rhamnoside-hexoside	28.1	350	C_27_H_30_O_15_	593	447(2.0), 285(100)		[28]
**11**	Luteolin 7-*O*-glucoside *	28.5	340	C_21_H_19_O_11_	447	285(100)		[29,27]
**14**	Apigenin hexoside	33.1	340	C_21_H_20_O_10_	431	269(100)		[29,28]
**22**	Luteolin*	43.9	370	C_15_H_10_O_6_	285	255(76), 175(30)		[29]

* Analytes identified with authentic standards; ** in bold—ions subjected to MS^3^ fragmentation.

**Table 2 molecules-27-05144-t002:** Effects of OLE treatment on UVA-induced apoptosis of Hs68 cells.

Groups	Apoptotic Index
Control	0.11 ± 0.009
UVA	0.54 ± 0.103 ***
OLE5+UVA	0.28 ± 0.024 ***
OLE25+UVA	0.24 ± 0.013 ***
QU25+UVA	0.25 ± 0.009 ***

Data are shown as the means ± SD of six independent experiments. Pre-treatment with OLE decreased the cellular apoptosis compared to the UVA group (*** *p* < 0.001).

## Data Availability

Not applicable.

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
