# Peer review of "Valorization of the Photo-Protective Potential of the Phytochemically Standardized Olive (*Olea europaea* L.) Leaf Extract in UVA-Irradiated Human Skin Fibroblasts"

_molecules, 2022, doi:10.3390/molecules27165144_

Round 1

Reviewer 1 Report

This work is interesting since molecular mechanisms underlying the photoprotective effect of the standardized extract are shown. Phytochemicals are a matter of interest for the cosmetic industry since their worth biological effects and low or null toxic effects. However, the methodology must be improved, there are many missing data such as energy used in ESI to fragment (and it was used to compare with MS-literature spectra); how were prepared the samples used for experiments from ethanol-water extract?, only it was described that DMSO and DMEM were used but the volume and concentration were not described. In many results under the concentrations, 5 and 25 ug/mL were statistically nondifferent, how to justify the range used in the design experiment?. Why many results did not show data about a blanc (0 ug/mL OLE; did DMSO or EtOH concentration cause cellular damage). Can be comparable the results with regard to quercetin and gallic acid versus OLE extract, since solubility and cell uptake could be totally different? 

Author Response

Thank you very much for your comments, which helped us to revise the manuscript. Below are point-by-point responses to the Reviewers’ comments. In the revised version of the paper, we have made the following changes.

  • energy used in ESI to fragment (and it was used to compare with MS-literature spectra);

Regarding the Reviewer's query for the energy used in the ESI fragmentation, we would like to inform that the MS detector used in our studies was of ion trap type. This type of detector is designed for qualitative purposes only and does not enable adjustment and reading of the fragmentation energy for individual analytes. Instead, the system adjusts energy automatically, which is adequate for qualitative studies. For more clarity about the fragmentation parameters, we have added the following sentence in Section 3.3. Phytochemical Profiling (lines 526-528): "The MS/MS fragmentation amplitude was set to 1.0 V and automatically adjusted by the system in the range of 60–300% of this value."

  • how were prepared the samples used for experiments from ethanol-water extract?, only it was described that DMSO and DMEM were used but the volume and concentration were not described.

Thank you very much for these comments. Volumes and concentrations were described under Materials and Methods in the revised version of the manuscript (section 3.5. Cell Treatment and UVA Irradiation; lines 552-561) to clarify how the samples were prepared.

  • In many results under the concentrations, 5 and 25 ug/mL were statistically nondifferent, how to justify the range used in the design experiment?

Thank you very much for this comment. To be effective, the concentration of the photoprotective agent strongly depends on the UV radiation dose used in the experiment. Therefore, the concentration range of the extract was experimentally established at the beginning of the study based on the determination of the level of DNA damage and the viability of fibroblasts after UVA-irradiation (using the CCK-8 method), and then consistently used in further study. This information was included in the revised version of the manuscript (section 2.2. Prevention of UVA-induced death of fibroblasts by inhibition of the intrinsic and extrinsic apoptosis pathway; lines 297-301).

  • Why many results did not show data about a blanc (0 ug/mL OLE; did DMSO or EtOH concentration cause cellular damage).

We want to thank you very much for this comment. For each of the determined parameters, we made the same series of samples, samples of one series were irradiated and the other series were not. (This was described in section 3.5. Cell Treatment and UVA Irradiation; lines 552-570). Since the aim of the research was to assess the photoprotective effect of olive leaf extract, i.e. to determine to what extent preincubation of cells with the extract reduces a given harmful effect of UVA radiation, most of the results were presented as the difference between the exposed and unexposed samples (for a given extract concentration) in relation to this difference of the control (samples not treated with the extract; 0ug/ml OLE were taken as 100%). Nevertheless, the control values are given in the text under Results and Discussion, e.g. activity of caspase 3, 8 and 9 in control fibroblasts (lines 347-350), ROS level (lines 413-414), extent of the endogenous DNA damage for control (lines 397). DMSO (0.125% final concentration in the sample) did not cause any cellular damage. We used the commercial dry water/ethanol extract from the olive leaves that we dissolved in DMSO. We did not use ethanol to dissolve the extract in the study.  

  • Can be comparable the results with regard to quercetin and gallic acid versus OLE extract, since solubility and cell uptake could be totally different?

Thank you very much for these comments. Positive standards as pure substances always have different solubility or bioavailability parameters than extracts and are used as a benchmark for assessing the potency of an extract, not its properties. The use of pure substances as benchmarks is a standard procedure.  

Reviewer 2 Report

The present manuscripts describe the composition of. Olea europaea L.) leaf extract and some biological activities that indicate its potential use as active ingredient in cosmetic or pharmaceutical formulation. The authors are confident ins its use as a topical photoprotective, antioxidant, and anti-inflammatory agent.

The study is well designed and interesting. However, some aspects need to be clarified or improved.

·         Introduction. Lines 49-51. Reference 5 is not appropriate to support such assertion. Parrado et al., (2018) review the positive impact of oral dietary botanicals as complementary measures for photoprotection. Nothing related to skin adverse effects by the topical use of sunscreen is reported. Please, refer to articles that deal with (photo)irritant or/and (photo)allergic reactions and the use of habitual sunscreens. Moreover, the authors should include any reference dealing with the harmful effects of photoprotective agents that threat corals and other marine life.

·         Keratinocytes, the main component of the epidermis, together with melanocytes is the first line of defence against UV radiation. Authors should discuss the reason for using fibroblasts and not keratinocytes or melanocytes, because no photodamage directly related to dermal functions is reported (alterations in MMP secretion, for example).

Author Response

Thank you very much for your comments, which helped us to revise the manuscript. Below are point-by-point responses to the Reviewers’ comments. In the revised version of the paper, we have made the following changes.

  • Lines 49-51. Reference 5 is not appropriate to support such assertion. Parrado et al., (2018) review the positive impact of oral dietary botanicals as complementary measures for photoprotection. Nothing related to skin adverse effects by the topical use of sunscreen is reported. Please, refer to articles that deal with (photo)irritant or/and (photo)allergic reactions and the use of habitual sunscreens. Moreover, the authors should include any reference dealing with the harmful effects of photoprotective agents that threat corals and other marine life.

We want to thank you very much for this comment. Articles that deal with (photo)irritant or/and (photo)allergic reactions and the use of habitual sunscreens were cited – two articles were added (Introduction section, line 51). Also references dealing with the harmful effects of photoprotective agents that threat corals and other marine life were included (line 52).

  • Keratinocytes, the main component of the epidermis, together with melanocytes is the first line of defence against UV radiation. Authors should discuss the reason for using fibroblasts and not keratinocytes or melanocytes, because no photodamage directly related to dermal functions is reported (alterations in MMP secretion, for example).

Thank you very much for these comments. “It is believed that the UVA photons are only partly absorbed by the upper layers of skin and can penetrate in deepness into the dermis by contrast with UVB radiation which is mostly absorbed by several major cutaneous targets in the epidermis (Cadet et al, 2014). Fibroblasts, like keratinocytes and melanocytes, are model cells used in many in vitro studies on photoprotection mechanisms. In our study we used cultured fibroblasts, which are often used to assess the effects of UVA radiation and the photoprotective action of plant compounds. The study conducted by Battie et al suggested that dermal fibroblasts are more sensitive to UVA-induced oxidative stress than keratinocytes (Battie et al, 2014).” This explanation was added in section 2.2. Prevention of UVA-induced death of fibroblasts by inhibition of the intrinsic and extrinsic apoptosis pathway (lines 261-268).